# End-to-End Low-Light Enhancement for Object Detection with Learned Metadata from RAWs

**Xuelin Shen**[1][*]   **Haifeng Jiao**[1,3][*]   **Yitong Wang**[3]   **Yulin He** [1] **Wenhan Yang**[2] [†]

[1]Guangdong Laboratory of Artificial Intelligence and Digital Economy (SZ)
[2]Peng Cheng Laboratory
[3]College of Computer Science and Software Engineering, Shenzhen University
`shenxuelin@gml.ac.cn, jiaohaifeng@gml.ac.cn, wangyitong@gml.ac.cn`
`heyulin@gml.ac.cn, yangwh@pcl.ac.cn`

## Abstract

Although RAW images offer advantages over sRGB by avoiding ISP-induced distortion and preserving more information in low-light conditions, their widespread use is limited due to high storage costs, transmission burdens, and the need for significant architectural changes for downstream tasks. To address the issues, this paper explores a new raw-based machine vision paradigm, termed Compact RAW Metadata-guided Image Refinement (CRM-IR). In particular, we propose a Machine Vision-oriented Image Refinement (MV-IR) module that refines sRGB images to better suit machine vision preferences, guided by learned raw metadata. In detail, we propose a Cross-Modal Contextual Entropy (CMCE) network for raw metadata extraction and compression. It builds upon the latent representation and entropy modeling framework of learned image compression methods, and uniquely exploits the contextual correspondence between raw images and their sRGB counterparts to achieve more efficient and compact metadata representation. Additionally, we integrate priors derived from the ISP pipeline to simplify the refinement process, enabling a more efficient design. Such a design allows the CRM-IR to focus on extracting the most essential metadata from raw images to support downstream machine vision tasks, while remaining plug-and-play and fully compatible with existing imaging pipelines, without any changes to model architectures or ISP modules. We implement our CRM-IR scheme on various object detection networks, and extensive experiments under low-light conditions demonstrate that it can significantly improve performance with an additional bitrate cost of less than $10^{-3}$ bits per pixel. Code is available at `https://github.com/haifengjiao001/CRM-IR`.

## 1 Introduction

Raw images refer to unprocessed and uncompressed data captured directly from a camera's image sensor. Their retained sensor readings preserve linear scene radiance and full bit-depth precision. Raw images typically undergo in-camera Image Signal Processing (ISP) steps, including demosaicing, white balancing, gamma correction and compression, to remove perceptual redundancy and enhance visual appeal, ultimately producing the commonly seen sRGB images. However, as these ISP pipelines are primarily designed to satisfy human perceptual preferences, they often perform suboptimally in machine vision practice. Especially in low-light conditions, ISP pipelines apply nonlinear radiance amplification to enhance details and textures, inevitably amplifying the inherent noise introduced

---

[*]These authors contributed equally.
[†]Correspondence to: Wenhan Yang.

39th Conference on Neural Information Processing Systems (NeurIPS 2025).

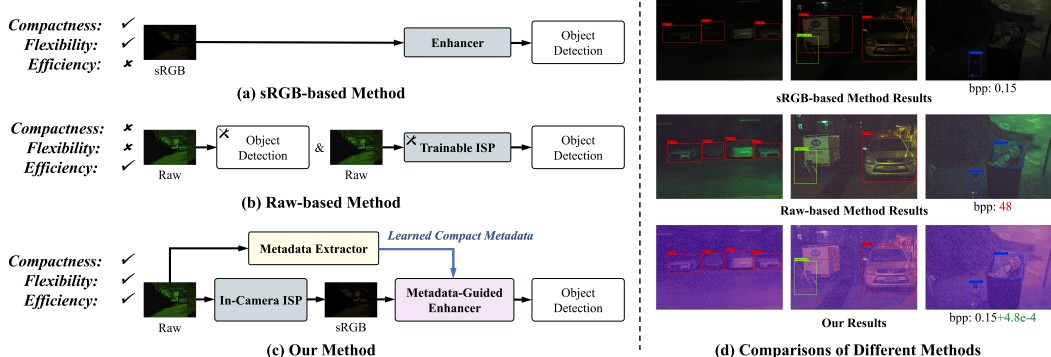

Figure 1: Comparison of existing low-light object detection methodologies: (a) sRGB-based methods, (b) raw-based methods and (c) the proposed CRM-IR. (d) visual results of representative approaches with respect to detection accuracy and transmission overhead, with the first and second rows reproduced from [16] and [17], respectively.

by physical factors during shooting, such as high ISO values, the use of flash and long exposure times [1, 2, 3, 4, 5]. Some studies apply low-light enhancement techniques to improve image quality under laser illumination, but these methods are constrained by the information loss during ISP processing [6, 7]. As a result, they can only adjust distributions and fail to fully recover many critical details. Moreover, these methods typically address human perceptual needs but do not sufficiently cater to the preferences or feature spaces required for machine vision, leaving this area of research still open.

Therefore, recent studies have revisited the full utilization of raw images, aiming to exploit their auxiliary and relatively clean visual information to improve machine vision applications [8, 9, 10]. Although the observed performance gains highlight the advantages of raw images over processed sRGB images in machine vision tasks, raw-based methods have shown progress but still face significant challenges that need to be addressed before they can be effectively applied in real applications. In particular, current approaches tend to take raw images as direct input [11, 12] and make substantial architectural changes to accommodate the different image format, whereas other studies [13, 14, 15] modify the in-camera ISP modules instead. These strategies pose flexibility issues as sRGB-based pipelines remain mainstream and perform well under normal lighting conditions. Moreover, the substantial storage and transmission resources requirements of raw images further hinder the practical adoption of raw-based solutions.

This paper introduces the Compact RAW Metadata-guided Image Refinement (CRM-IR) paradigm to tackle challenges in low-light machine vision, characterized by strong *flexibility* and *compactness*. The approach extracts only essential raw image information, boosting downstream models, and integrates seamlessly into existing vision pipelines without altering current architectures or ISP procedures, as shown in Fig. 1. At its core, CRM-IR features a lightweight Image Refinement (MV-IR) module, which uses raw metadata as external conditions for pixel-level modifications to processed sRGB images *flexibly*. Through joint end-to-end training with metadata extraction and downstream networks, the MV-IR module flexibly adapts to machine vision needs. To ensure *compactness*, a raw-metadata encoder, based on Learned Image Compression (LIC), is introduced. This encoder extracts and compresses raw metadata efficiently, utilizing a cross-modal contextual entropy coding strategy that leverages semantic correspondence between the sRGB image and raw data for effective compression. Additionally, we present the Raw in Dark (RID) dataset, containing 500 annotated RAW sensor pairs from low-light daily scenes, further advancing RAW-based object detection. Extensive experiments demonstrate that CRM-IR achieves superior performance compared to existing methods, with minimal metadata transmission of less than 0.001 bits per pixel.

Our contributions are summarized as follows:

- We propose a novel raw-based machine vision paradigm that extracts only the most essential information from raw images to guide machine-oriented sRGB image refinement. This

design enables seamless integration as a plug-in within existing sRGB-based vision pipelines while maintaining minimal storage and transmission overhead.

- We introduce a novel raw-metadata encoder that effectively leverages the cross-modal contextual information between processed sRGB and raw images. This enables the proposed scheme to transmit only a minimal amount of raw metadata while significantly enhancing downstream performance.

- To advance RAW-based machine vision, we construct Raw in Dark (RID)—a diverse, large-scale dataset of 500 annotated RAW-sRGB pairs captured in real-world low-light scenarios across 8 object categories. RID fills key gaps in existing open-source datasets and serves as a strong benchmark for evaluating generalization in RAW-guided object detection through cross-dataset validation.

## 2 Related Work

### 2.1 Raw-based Machine Vision

The last decade has witnessed substantial progress in machine vision techniques, markedly improving scene-understanding accuracy [18, 19, 20, 21, 22] and inference speed [23, 24, 25, 26] and enabling widespread adoption in real-world applications. Nevertheless, robust performance under low-light conditions remains elusive. Dim illumination limits photon counts and produces a low signal-to-noise ratio [3, 4], while compensatory measures such as high-ISO amplification or extended exposure add sensor noise and motion blur [5]; together these factors shift the input distribution away from the priors learned during training and sharply reduce model accuracy. Early studies inserted a preprocessing stage to enhance semantic information before inference [27, 28], yet this approach did not achieve satisfactory performance and constrained the models' generalizability and scalability across real-world benchmarks and diverse tasks. Consequently, recent research has turned to raw images, seeking to harness their abundant unprocessed visual information to improve both image enhancement and downstream machine-vision performance. For instance, some studies [11, 12] adopt a straightforward strategy by training or fine-tuning downstream models directly on raw images, whereas others [13, 14, 15] focus on the ISP pipeline and employ differentiable image signal processors to generate sRGB images tailored for machine-vision tasks. Although these raw-based methods deliver substantial gains, they require full access to raw data, which hampers deployment in edge-to-cloud scenarios where transmitting full-resolution raw images is impractical.

### 2.2 Learned Image Compression

In recent years, Learned Image Compression (LIC) has progressed rapidly in step with deep-learning breakthroughs. In particular, Ballé *et al.* [29] spearheaded this shift by replacing handcrafted transforms, quantizers and entropy coders with a single, fully trainable pipeline. They later augmented their framework with a hyperprior that conditions each latent on auxiliary hyper-latents, markedly improving the rate-distortion trade-off over factorized-prior baselines [30]. Follow-up studies refined the entropy model by exploiting contextual cues: local part-of-image context [31], context spanning the entire image [32], checkerboard inference patterns [33], and channel-wise context [34, 35] have all been employed to boost accuracy or reduce computation. A recent exemplar, MLIC++ by Jiang *et al.* [36], fuses local, global and channel contexts in a multi-reference entropy model and already surpasses the latest coding standard, Versatile Video Coding (VVC) [37]. Meanwhile, other LIC works tend to enhance the analysis–synthesis backbone itself by adopting residual network [38], invertible network [39] and Swin-Transformer [40, 41], yielding richer latent representations and further gains in compression.

## 3 Method

### 3.1 Motivation and Overview

To address the limitations of existing raw-based methods in terms of flexibility and compactness, we propose a novel framework called Compact RAW Metadata-guided Image Refinement (CRM-IR). CRM-IR aims to fully leverage raw images to enhance downstream machine vision tasks while

maintaining a lightweight and pluggable structure compatible with existing vision pipelines. The overall framework is illustrated in Fig. 2, consisting of three key components.

**1) Raw Metadata Extraction**. Let $x_r$ denote the input raw image and $x_s$ represent its sRGB counterpart. At the imaging stage, we employ a raw metadata encoder $G(\cdot; \boldsymbol{\omega})$ parameterized by $\boldsymbol{\omega}$ that takes as input $x_r$ while being conditioned on $x_s$, to identify and extract the raw metadata $y$,

$$y = G(x_s, x_r; \boldsymbol{\omega}). \tag{1}$$

**2) Metadata Coding**. Subsequently, a hyperprior-based entropy encoder $E(\cdot, \boldsymbol{\theta})$ parameterized by $\boldsymbol{\theta}$ is adopted to capture the statistical property of $\hat{y}$ under a multivariate Gaussian distribution, while simultaneously estimating and constraining its entropy, denoted as $E(y; \boldsymbol{\theta})$ for simplicity, which will be elaborated later.

**3) Image Refinement for Vision Tasks**. At the application end, an image refinement model $M(\cdot; \boldsymbol{\phi})$ is incorporated, being responsible for adapting $x_s$ to align with the requirements of downstream machine-vision tasks guided by $y$,

$$\widehat{x_s} = M(x_s, y; \boldsymbol{\phi}), \tag{2}$$

where $\widehat{x_s}$ is the refined sRGB image, $\boldsymbol{\phi}$ denotes the model parameter.

By feeding the $\widehat{x_s}$ to the downstream machine vision models $A(\cdot, \boldsymbol{\psi})$, the entire CRM-IR scheme is capable of end-to-end training under the following constraint,

$$\bar{\boldsymbol{\theta}}, \bar{\boldsymbol{\phi}}, \bar{\boldsymbol{\psi}}, \bar{\boldsymbol{\omega}} = \underset{(\phi, \psi, \theta, \omega)}{\arg\min} \sum_{(x_r, x_s, z) \in D} \lambda \cdot \mathcal{L}_A(z, A(M(x_s, G(x_s, x_r; \boldsymbol{\omega}); \boldsymbol{\phi}); \boldsymbol{\psi})) + E(y; \boldsymbol{\theta}), \tag{3}$$

where $D$ denotes the training set, consisting of the raw image $x_r$, sRGB image $x_s$ and annotation $z$ for machine vision task. $\mathcal{L}_A$ measures the task performance, while $\lambda$ is the Lagrange parameter to balance the bits cost and downstream task performance.

To meet the dual goals of *compactness* and *flexibility*, we incorporate the following designs:

- **Compactness**: We introduce a Cross-Modal Contextual Entropy Encoder (CMCE). The raw metadata must comprise only essential information from $x_r$ while remaining independent of $x_s$ with respect to machine vision requirements. CMCE employs the sRGB image $x_s$ as a contextual prior during raw metadata compression. This approach effectively captures the inter-redundancy between the raw and the sRGB images.

- **Flexibility**: We design a lightweight Machine Vision-oriented Image Refinement (MV-IR) module, which modifies $x_s$ at the pixel level under the guidance of metadata $y$. MV-IR introduces no changes to the image format, allowing seamless integration into existing pipelines without altering downstream architectures.

Detailed descriptions of the proposed CMCE and MV-IR are provided in the following subsections.

## 3.2 Cross-modal Contextual Entropy Encoder

In the field of LIC, context-based entropy modeling involves utilizing surrounding contexts, *i.e.*, information from already encoded latent elements, to dynamically estimate the probability distribution for encoding subsequent elements. This approach leverages complex correlations in the input images to improve compression efficiency. Motivated by this, we utilize the cross-modal dependency between the sRGB image $x_s$ and raw image $x_r$, both available at the imaging end, to further enhance the compression process. In particular, the $x_s$ would be first concatenated with the $x_r$ while ensure that its latent elements are leveraged as contextual priors for estimating the probability distribution of the latent representation of $x_r$. After obtaining their joint latent representation $y$ via Eqn. (1), a hyperencoder is introduced to extract side information $z = h_a(y)$, which captures the spatial dependencies among the elements of $y$.

In coding practice, $y$ and $z$ are typically passed through a uniform quantization process to obtain their integer forms $\hat{y}$ and $\hat{z}$ for compression purposes. During training, this quantization is approximated using uniform noise $\mathcal{U} \sim (-\frac{1}{2}, \frac{1}{2})$. As such, the actual rate estimation can be formulated as,

$$\mathcal{R}(\hat{y}) + \mathcal{R}(\hat{z}) = \mathbb{E}[-log_2(p_{\hat{y}|\hat{z}}(\hat{y}|\hat{z}))] + \mathbb{E}[-log_2(p_{\hat{z}}(\hat{z}))]. \tag{4}$$

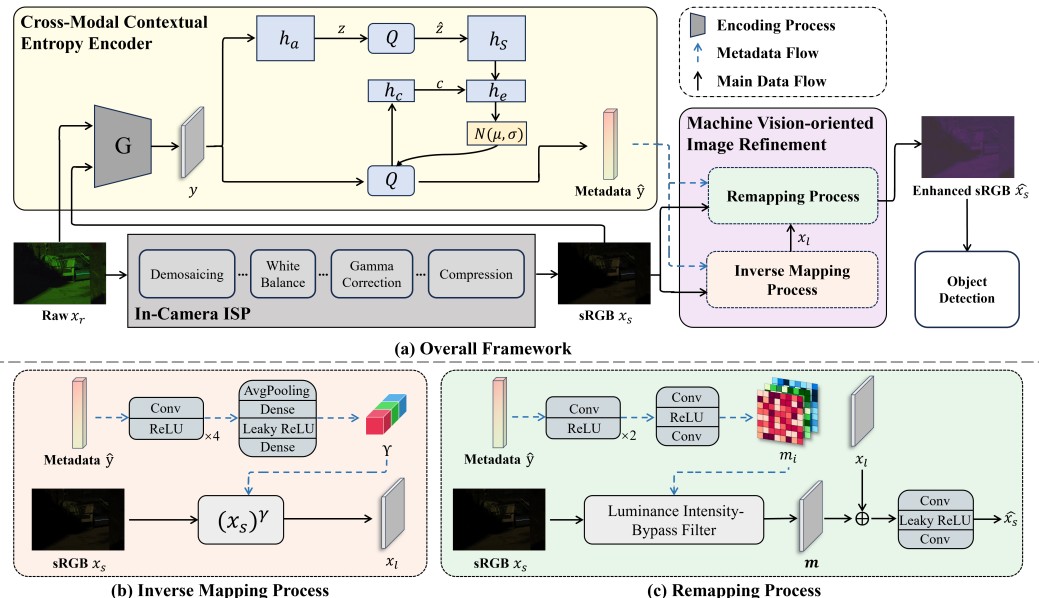

Figure 2: (a) Overall framework of the proposed CRM-IR framework. (b) and (c) details the inverse mapping and the remapping process of the proposed MV-IR module, respectively.

To obtain $p_{\hat{y}|\hat{z}}(\hat{y}|\hat{z})$, a multivariate Gaussian prior is introduced, parameterized by a hyperdecoder $h_s(\cdot)$,

$$p_{\hat{y}|\hat{z}}(\hat{y}|\hat{z}) \sim \mathcal{N}(\boldsymbol{\mu}, \boldsymbol{\sigma}), [\boldsymbol{\mu}, \boldsymbol{\sigma}] \leftarrow h_s[\hat{z}; \theta_s], \tag{5}$$

where the $\theta_s$ denotes the parameters of $h_s(\cdot)$. Within this process, an autoregressive context model $h_c$ is employed to capture the contextual information within $\hat{y}$. It utilizes both the side information $\hat{z}$ and already processed and the already processed portion of $\hat{y}$ to predict the distribution of the remaining elements. Accordingly, the causal context $c_i$ for a given latent element is obtained via $c_i = h_c(\hat{\mathbf{y}}_{<i}; \theta_c)$, where $\hat{\mathbf{y}}_{<i}$ denotes the causal context of $y_i$. The prediction of the Gaussian parameters can be roughly formulated as a function involving the hyper-decoder, context model and entropy parameter network,

$$\mu_i, \sigma_i = h_e(h_s(\hat{z}; \theta_s), c_i; \theta_e), \tag{6}$$

where $h_e(\cdot; \theta_e)$ denotes the entropy parameter network with parameter $\theta_e$. Thus, we obtained

$$p_{\hat{y}|\hat{z}}(\hat{y}|\hat{z}) = \prod_i \left( \mathcal{N}(\mu_i, \sigma_i) * \mathcal{U}(-\frac{1}{2}, \frac{1}{2}) \right)(\hat{y}_i). \tag{7}$$

As for the entropy coding of the side information $\hat{z}$, *i.e.,* the estimation of $\mathbb{E}[-log_2(p_{\hat{z}}(\hat{z}))]$ in Eqn. (4) since no hyperprior is available for this process, a simple factorized density model is employed. This model is directly adopted from existing LIC works and is not elaborated upon here.

### 3.3 Machine Vision-oriented Image Refinement based on Raw Metadata

As revealed by existing raw-based works [1], within the ISP process, white balancing and gamma correction are the main modules that negatively affect downstream machine vision tasks, as both involve nonlinear mapping of pixel luminance to a discrete domain aligned with human perceptual preferences. Therefore, the main idea of the proposed MV-IR is to first convert the sRGB image back to a linear space, guided by the raw metadata. Subsequently, an additional remapping process is introduced to enhance details and structures in low-light regions, specifically tailored to machine vision requirements.

In particular, during the inverse remapping process, to simplify the task, we introduce an additional prior based on gamma correction, *i.e.,* we aim to predict an image-wise gamma correction parameter that is also aligned with the ISP process. Thus, the inverse mapping process can be formulated by,

$$x_l = (x_s)^\gamma, \gamma = F_{im}(\hat{y}), \tag{8}$$

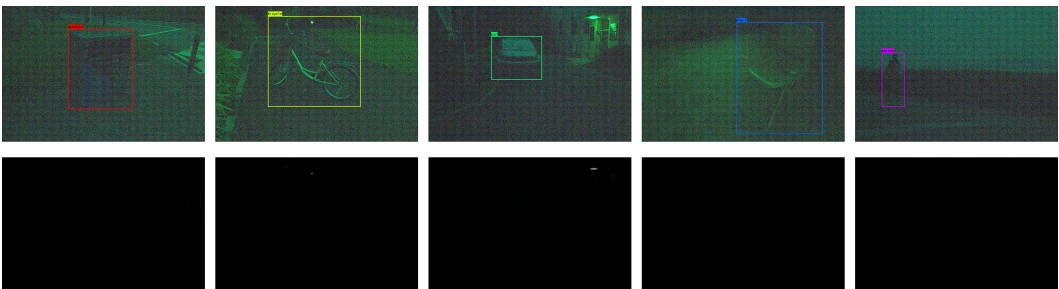

Figure 3: Samples from the RID dataset: The first row shows RAW images and the second row shows sRGB images processed by the camera's internal ISP.

where $x_l$ denotes the linear representation of $x_s$, and $F_{im}(\cdot)$ is responsible for predicting the gamma parameter. This prediction is straightforward with the assistance of the raw metadata and can be efficiently implemented using only a few convolutional layers, as shown in Fig. 2 (b).

As for the remapping process illustrated in Fig. 2 (c), $F_{rm}(\cdot)$ is employed with the aim of learning a pixel-wise modulation map $\mathbf{m}$, conditional on $x_s$ and $\hat{y}$. Considering that our method is specifically designed for low-light scenarios, we introduce a luminance region-wise learning strategy, *i.e.*, explicitly learning distinct modulation maps $[m_1, m_2]$ corresponding to low-light and normal-light regions.

$$m_i = F_{rm}(x_s, \hat{y}), i \in (1, 2), \tag{9}$$

where $F_{rm}(\cdot)$ is for the modulation map prediction. They would be subsequently merged to $\mathbf{m}$ via through luminance intensity-bypass filtering operations.

$$\mathbf{m} = \sum_{i=1}^{2} m_i \cdot \delta_i(x_s), \tag{10}$$

where $\delta_1(\cdot)$ and $\delta_2(\cdot)$ denote pixel-wise bypass filters that selectively pass the corresponding elements of the learned maps when the pixel intensities fall within the low-light and normal-light regions, respectively. Afterward, the modulation map $\mathbf{m}$ is applied to the linear representation $x_l$ via summation and subsequently fed into lightweight fusion layers $F_f(\cdot)$ for further adjustment, producing the final output.

$$\widehat{x_s} = F_f(\mathbf{m} + x_l). \tag{11}$$

### 3.4 Raw in Dark(RID) Dataset

This subsection introduces the RID dataset to enable cross-dataset generalization validation, given the scarcity of open-source real–world benchmarks. Compared with the widely employed raw-based object detection dataset LOD [42], RID broadens the range of scenes and object categories and places particular emphasis on images captured under extremely low-light conditions. As for the source image collection, a Canon EOS80D was used to capture 500 paired samples, each consisting of a RAW image and its sRGB-JPEG counterpart. We selected scenes that span a range of everyday environments, including dimly lit underground parking garages, nighttime roadsides and poorly illuminated indoor rooms, all characterized by deep shadows and severely limited visibility. We selected ISO settings of 50 and 100 and exposure times of 1/125s and 1/250s to replicate the photon-starved conditions typical of extremely dark surveillance scenarios.

Regarding the annotations, professional annotators are employed to create accurate instance-level labels for widely studied classes, *i.e.*, car, bicycle, chair, to support cross-task generalization validation, and further categories including person, zebra crossing, emergency exit sign, fire extinguisher and dustbin are included to extend the dataset's coverage. Representative examples are exhibited in Fig. 3. It is worth noting that RID is an ongoing project. We plan to further expand the dataset by employing various shooting devices, including different cameras and smartphones, with the goal of investigating the influence of diverse ISP procedures. Additionally, we will include more annotations for other critical machine vision tasks, such as semantic segmentation and instance segmentation, aiming to support the broader machine vision research community.

Table 1: Quantitative comparison results on the YOLOv3 backbone. The bold and underlined font indicate the best and second-best results, respectively. For reference, JPEG-format sRGB images have an average bpp of 0.15, whereas raw images require 48 bpp.

| Method | In-Dataset Val. (LOD) | | | | Cross-Dataset Val. (RID) | | | |
|---|---|---|---|---|---|---|---|---|
| | mAP | mAP50 | mAP75 | bpp | mAP | mAP50 | mAP75 | bpp |
| YOLOv3 | 40.00 | 67.67 | 43.44 | - | 45.67 | 65.80 | 63.75 | - |
| Zero-DCE | 41.19 | 67.79 | 45.79 | - | 45.26 | 64.38 | 62.76 | - |
| KinD | 41.22 | 67.03 | 44.60 | - | 44.37 | 63.76 | 58.55 | - |
| YOLA | 41.00 | 68.49 | 45.87 | - | 38.29 | 60.22 | 46.29 | - |
| RAOD | 41.60 | **70.49** | 42.29 | - | 39.74 | **68.49** | 41.36 | - |
| Ours | **42.14** | 69.11 | **46.02** | 4.88e-4 | **51.72** | 65.81 | **63.85** | 4.86e-4 |

# 4 Experiments

At the experimental stage, we implemented our CRM-IR scheme on a set of object detection models and conducted extensive comparisons with other low-light object detection paradigms, including both sRGB-based and raw-based methods, to demonstrate the superiority of our approach. Moreover, comprehensive ablation studies were performed to validate the effectiveness of the proposed CMCE and MV-IR modules.

## 4.1 Experimental Settings

**Datasets.** Low-light Object-Detection (LOD) dataset [42] is employed as our benchmark, which contains 2230 paired RAW and sRGB-JPEG format image pairs collected by a Canon EOS 5D Mark IV camera, covering both low-light and normal daylight scenes, where only the low-light parts are selected in our experiments. In particular, 1,784 training pairs and 446 test pairs are selected for model training and evaluation, respectively. These images contain a total of 9,726 labeled instances spanning 8 common object classes: car, motorbike, bicycle, chair, dining table, bottle, TV monitor and bus. In addition, our proposed RID dataset is employed to perform cross-dataset testing, aiming to evaluate the generalization capability of the employed methods.

**Baselines.** Three milestone object detection models are employed as baseline models, including YOLOv3 [43], Faster R-CNN [44] and CenterNet [21]. Moreover, four state-of-the-art low-light object detection schemes are evaluated, including three sRGB-based methods: Zero-DCE [16], KinD [45] and YOLA [46], all of which follow a similar pipeline that first enhances low-light images before feeding them into downstream object detection models. Moreover, one raw-based method RAOD [17] is also adopted. It follows a similar enhancement-based pipeline by leveraging raw images to guide the enhancement process, but it overlooks the transmission cost of raw data. For a fair comparison, all competing schemes are implemented on the same baseline models as our method and trained from scratch using the same dataset.

**Evaluation Criterion.** We adopt the commonly used mean average precision regarding mAP, mAP50 and mAP75 to measure the downstream task performance. Moreover, to demonstrate the compactness of our method, we report the bpp as an indicator of storage or transmission resource requirements across different methods. For the employed methods, we report the bpp of sRGB images and raw images for sRGB-based and raw-based approaches, respectively. In contrast, for our method, we additionally report the bpp of the raw metadata alongside the sRGB image.

**Implementation Details.** During training, data augmentation strategies are employed, including random horizontal flips and random scale jitter during resizing. All models were trained for 300 epochs using the Adam optimizer [47]. A linear scaling learning rate with a cosine decay schedule was employed, starting from an initial learning rate of 5e-4. The weight decay was set to 0, momentum was 0.9 and the training batch size was set to 8. During both training and testing, all input images were resized to $512 \times 512$. All experiments were conducted using PyTorch on an NVIDIA RTX 6000 Ada Generation GPU with 48 GB of memory.

Table 2: Quantitative comparison results on the Faster R-CNN backbone. The bold and underlined font indicate the best and second-best results, respectively.

| Method | In-Dataset Val. (LOD) | | | | Cross-Dataset Val. (RID) | | | |
|---|---|---|---|---|---|---|---|---|
| | mAP | mAP50 | mAP75 | bpp | mAP | mAP50 | mAP75 | bpp |
| Faster R-CNN | 39.96 | 67.12 | 41.74 | - | 37.23 | 63.79 | 39.30 | - |
| Zero-DCE | 40.50 | 67.94 | 42.56 | - | 37.73 | 64.57 | 40.07 | - |
| KinD | 39.83 | 66.86 | 42.23 | - | 37.11 | 63.54 | 39.76 | - |
| YOLA | 40.31 | 68.22 | 42.31 | - | 37.55 | 64.83 | 39.84 | - |
| RAOD | 41.32 | **69.79** | 43.14 | - | 38.50 | **66.33** | 40.62 | - |
| Ours | **41.75** | 68.49 | **43.94** | 3.26e-4 | **38.89** | 65.09 | **41.37** | 3.23e-4 |

Table 3: Quantitative comparison results on the CenterNet backbone. The bold and underlined font indicate the best and second-best results, respectively.

| Method | In-Dataset Val. (LOD) | | | | Cross-Dataset Val. (RID) | | | |
|---|---|---|---|---|---|---|---|---|
| | mAP | mAP50 | mAP75 | bpp | mAP | mAP50 | mAP75 | bpp |
| CenterNet | 40.41 | 65.36 | 42.60 | - | 40.33 | 62.63 | 44.67 | - |
| Zero-DCE | 41.44 | 65.85 | **44.90** | - | 40.30 | 62.64 | 44.53 | - |
| KinD | 40.10 | 62.49 | 44.23 | - | 40.28 | 62.48 | 44.52 | - |
| YOLA | 41.74 | 66.18 | 44.55 | - | 42.04 | 66.48 | 44.85 | - |
| RAOD | **42.89** | **69.25** | 43.94 | - | **43.45** | **70.28** | 43.22 | - |
| Ours | 42.05 | 68.09 | 44.52 | 1.38e-3 | 42.31 | 68.28 | **44.91** | 1.39e-3 |

## 4.2 Experimental Results

The quantitative evaluation results regarding the baseline models of YOLOv3, Faster R-CNN and CenterNet are established in Table 1, Table 2, Table 3, respectively. First, the effectiveness of the proposed method can be easily observed comparing with the baseline models, as leading to an average mAP improvement of 2.14%, 1.79% and 1.64% regarding YOLOv3, Faster R-CNN and Centernet, respectively. Considering bitrate performance, our method introduces only 4.88e-4, 3.26e-4 and 1.38e-3 bpp of raw-metadata overhead for the three baseline models, respectively. Relative to the original sRGB bitrate of 0.15 bpp, this increase is under 0.5 percent and therefore negligible in practice, thanks to the CMCE module's ability to capture and exploit the contextual correspondence between each raw image and its sRGB counterpart.

Compared with state-of-the-art sRGB-based methods Zero-DCE, KinD and YOLA, our scheme achieves an average mAP improvement of about 0.94%, 1.60% and 0.96%, confirming that leveraging raw-image information effectively overcomes the limitations of conventional ISP processing. Compared with the raw-based method RAOD, our approach achieves comparable performance while transmitting only a compact metadata stream instead of the entire raw image. RAOD benefits from full access to the 48 bpp raw data, yet this requirement imposes a prohibitive storage and transmission load, making the scheme impractical for many real-world deployments. Cross-dataset evaluations confirm the strong generalization capability of our method, as it maintains nearly the same level of object-detection accuracy in unseen scenarios.

Fig. 4 provides a set of intuitive comparisons. Things have to be mentioned that, distinct visualization strategies were adopted for a clear representation. For the Ground Truth, detection bounding boxes are visualized directly on the original low-light input. For the other approaches that follow the "*enhancement-then-detection*" pipeline, detection results are shown on their corresponding enhanced images. Fig. 4 demonstrates that low light sRGB images contain considerable environmental noise that the enhancement stage cannot fully remove, causing frequent mis-detections. By contrast, incorporating raw information reduces noise in the enhance version and substantially improves downstream detection accuracy.

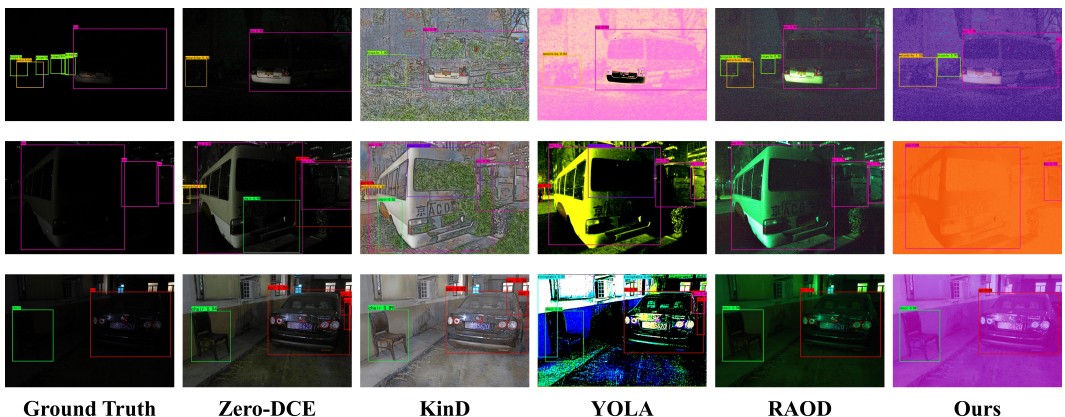

| Ground Truth | Zero-DCE | KinD | YOLA | RAOD | Ours |

Figure 4: Visualization of object-detection results: the first, second and third rows correspond to methods based on YOLOv3, Faster R-CNN and CenterNet, respectively.

## 4.3 Ablation Studies

To evaluate the distinct contributions of our proposed modules and the utility of leveraging raw image data, we performed a detailed ablation study on the LOD datasets. We compare our full model against several ablated variants:

$w/o$ **CMCE:** In this configuration, the metadata extraction and compression stages are disabled, so no raw metadata are passed to the MV-IR module. The MV-IR remains active but is driven by a non-informative placeholder implemented as a zero tensor.

$w/o$ **MV-IR:** We retain the CMCE module but employ the enhancement module with simple concatenation, where the extracted metadata and the sRGB image features are combined via concatenation.

The ablation results are presented in Table 4, where the effectiveness of each specially designed module is evident. As shown, excluding the raw metadata yields an average mAP50 decrease of about $0.58\%$, $2.99\%$ and $2.24\%$ regarding YOLOv3, Faster R-CNN and CenterNet, respectively. Meanwhile, omitting the MV-IR module results in an overall $1.91\%$, $3.26\%$ and $1.77\%$ decrease in YOLOv3, Faster R-CNN and CenterNet, respectively.

Table 4: Ablation Study Results of CMCE and MV-IR modules

| Detector | CMCE | MV-IR | mAP | mAP50 | mAP75 |
|---|---|---|---|---|---|
| YOLOV3 | ✗ | ✗ | 40.00 | 67.67 | 43.44 |
| | ✗ | ✓ | 41.06 | 68.53 | 44.78 |
| | ✓ | ✗ | 41.19 | 67.20 | 45.21 |
| | ✓ | ✓ | 42.14 | 69.11 | 46.02 |
| Faster R-CNN | ✗ | ✗ | 39.96 | 67.12 | 41.74 |
| | ✗ | ✓ | 39.65 | 65.85 | 42.40 |
| | ✓ | ✗ | 38.84 | 65.23 | 41.13 |
| | ✓ | ✓ | 41.75 | 68.49 | 43.94 |
| CenterNet | ✗ | ✗ | 40.41 | 65.36 | 42.60 |
| | ✗ | ✓ | 41.16 | 65.85 | 43.67 |
| | ✓ | ✗ | 41.64 | 66.32 | 44.25 |
| | ✓ | ✓ | 42.05 | 68.09 | 44.52 |

## 5 Conclusion

To conclude, this paper explores a new raw-based object-detection paradigm called Raw Metadata-guided Image Refinement (CRM-IR). Compared with existing raw-based approaches,

CRM-IR is characterized by its flexibility and compactness: it integrates seamlessly into current machine vision pipelines while explicitly accounting for the transmission and storage overhead of raw information. For flexibility, we introduce a Machine Vision-oriented Image Refinement (MV-IR) module that adjusts sRGB images to better match machine vision preferences, functioning as a standalone preprocessing step without altering network architectures or in-camera ISP modules. CRM-IR further leverages raw metadata collected at the imaging stage; by extracting only the essential raw information and using the paired sRGB image as contextual prior, it delivers significant gains in downstream object-detection accuracy with negligible additional bitrate.

## Acknowledgments

This work was in part by the Interdisciplinary Frontier Research Project of PCL under Grant 2025QYB013, in part by the Major Key Project of PCL (PCL2025A03), in part by Guangdong Basic and Applied Basic Research Foundation under Grant 2024A1515010454, in part by the Open Research Fund from Guangdong Laboratory of Artificial Intelligence and Digital Economy (SZ) under Grant No. GML-KF-24-27, in part by the Natural Science Foundation of Guangdong Province under Grant 2023A1515011667.

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
