# OpenReview forum: "End-to-End Low-Light Enhancement for Object Detection with Learned Metadata from RAWs"
_NeurIPS.cc/2025/Conference — NeurIPS 2025 poster_

### Official Review · Reviewer_aGYu · 2025-06-07

**Clarity:** 2
**Significance:** 2
**Originality:** 3
**Rating:** 4
**Confidence:** 4

**Summary:**

This paper introduces a novel paradigm for low-light object detection called Compact RAW Metadata-guided Image Refinement (CRM-IR). The core idea is to leverage information from RAW camera sensor data without incurring the high storage and transmission costs typically associated with it. Instead of processing the full RAW image, the proposed method extracts a small amount of essential "metadata" from the RAW data. This metadata is then used by a lightweight module to refine a standard, compressed sRGB image, making it better suited for machine vision tasks.

**Questions:**

- What is the computational overhead? Could you please provide details on the increase in inference time (or decrease in FPS) when adding the CRM-IR modules to the baseline object detectors? This is a critical factor for real-world deployment.
- The method learns to refine an sRGB image based on metadata from its corresponding RAW version. Since the sRGB image is the output of a specific, often proprietary, in-camera Image Signal Processor (ISP), how well does the trained model generalize to images from different cameras with different ISPs? Have you performed any experiments to test this robustness?
-  And also above weakness

**Ethical Concerns:**

["NO or VERY MINOR ethics concerns only"]

**Final Justification:**

The authors partially addressed my concerns, but it doesn't fully validate the accuracy of the detection. However, the article's contribution to the dataset and innovative approach can be considered a significant contribution.

**Limitations:**

Yes

**Quality:**

2

**Strengths And Weaknesses:**

Strengths: The proposed solution is highly practical as it circumvents the high costs of transmitting and storing RAW images, which is a major bottleneck for many real-world applications, especially in edge-to-cloud scenarios. The plug-and-play nature of the proposed module makes it easily integrable into existing systems without major architectural changes.

The work suffers from several significant weaknesses that should be addressed:

1. The Contribution of the Proposed RID Dataset is Limited. The authors introduce the Raw in Dark (RID) dataset for cross-dataset generalization validation. However, the overall contribution and novelty of this dataset appear to be limited considering existing LOD . The RID dataset follows the existing paradigm of paired RAW and sRGB images, which is structurally identical to the benchmark LOD dataset. It does not introduce a new format or annotation methodology that would address fundamental challenges in the field. With only 500 paired samples, the dataset is considerably smaller than the LOD benchmark, which contains 2,230 pairs. Although the dataset focuses on more extreme low-light scenes and expands the number of object categories, this represents an incremental addition to existing resources rather than a foundational or groundbreaking contribution.

2. A critical weakness is the paper's failure to address the method's generalization capability across heterogeneous camera sensors, especially those with different Color Filter Arrays (CFAs). The method operates directly on RAW data, and all experiments were conducted on two Canon camera models that use a standard Bayer (RGGB) CFA. It is highly probable that the model is learning priors that are specific to this RGGB pattern. The paper provides no discussion or evaluation of how the model would perform on RAW data from sensors with different CFAs, such as RYYB or X-Trans, which are common in smartphones and other camera systems. A model trained on RGGB data would likely face a severe performance degradation when applied to a different RAW structure due to the fundamental mismatch in color channel information and spatial layout. This unaddressed dependency on specific hardware severely limits the method's claimed "plug-and-play" applicability.

3. The paper's evaluation against the "enhancement-then-detection" paradigm is insufficient, as the selection of baseline methods is too narrow. The authors only select three sRGB enhancement algorithms (Zero-DCE, KinD, and YOLA) to form the two-stage baseline. The field of Low-Light Image Enhancement contains many high-performing algorithms that could form a much more competitive baseline, such as the pioneering RAW-based enhancer SID (which is cited but not used for comparison) or more recent state-of-the-art models like Rawformer. By failing to compare against a more comprehensive and challenging set of SOTA enhancement methods, the paper's claim that its end-to-end approach is superior to two-stage pipelines is not robustly supported. It is possible that a pipeline combining a top-tier LLE algorithm with a detector could achieve comparable or even better performance.

---

> ### Author Rebuttal · Authors · 2025-07-31
>
> First of all, thank you very much for your professional and thoughtful review. We will do our best to incorporate your suggestions to further improve the quality of our manuscript. Regarding your concerns, we hope the following point-by-point responses adequately address them.
>
> Q1. Concerns regarding the contribution of the constructed dataset
>
> Ans.: Thank you for your comments. First, we would like to emphasize that the core contribution of our work lies in efficiently extracting the minimal yet essential RAW information required by downstream machine vision tasks, with careful consideration of transmission overhead. The dataset serves as just one component supporting this broader objective.
>
> Regarding the dataset, our primary motivation was to create a resource for cross-dataset validation, which is crucial not only for our method but for all RAW-based object detection approaches. To the best of our knowledge, at the time of submission, only one other real-night RAW dataset with three overlapping object categories existed—ROD (*Toward RAW Object Detection : A New Benchmark and A New Model*, CVPR 2023). However, it only contains RAW images with corresponding annotations (no sRGB counterpart), preventing it from rigorous cross dataset evaluation. In this context, our constructed dataset fills an immediate gap and provides a meaningful contribution to the low light research community.
>
> Q2. Provides discussion or evaluation on RAW data from sensors with different CFAs
>
> Ans.: Thanks for your constructive and insightful suggestions. First, we agree that camera-specific details, such as the CFA pattern, are indispensable for fidelity-oriented RAW reconstruction tasks. However, our objective is different: we extract only the metadata that downstream machine vision models require but cannot obtain from sRGB images, and we deliberately omit fidelity or perceptual supervision to minimize the amount of RAW information that needs to be transmitted. Accordingly, the MV IR module applies an inverse gamma step followed by a learnable remapping to recover the information suppressed by gamma correction and compression, which depend only minimally on the camera model. In this way, our method achieves satisfactory performance with little to no reliance on the full set of camera-dependent parameters.
>
> However, at the current stage, we are not able to obtain such an open source dataset to conduct a rigorous cross dataset evaluation, and we appreciate your further suggestions.
>
> Moreover, we sincerely appreciate your constructive suggestions. As noted in our paper, the construction of the RID dataset is an ongoing project, and we plan to incorporate additional subsets captured with diverse smartphones (e.g., HUAWEI and iPhone) to further contribute to the broader machine vision community.
>
>
> Q3. Comparing with other RAW-based enhancement methods.
>
> Ans.: Thanks a lot for your suggestions. First, we acknowledge that recent RAW-based methods have demonstrated overwhelming advantages over sRGB-based approaches; we will incorporate more RAW-based methods in the related-works section and experiments of our manuscript. In fact, we have already included the cutting-edged RAW-based enhancement method RAOD (CVPR 2023) which is particular designed for object detection tasks, and our approach achieves comparable performance while offering unique advantages in terms of transmission overhead.
>
> Meanwhile, we would like to emphasize that in comparisons with RAOD and other RAW-based methods, our focus is not solely on achieving higher performance, which is unrealistic given that those approaches have full access to the entire RAW data. Our primary objective is to minimize the transmission and storage requirements of RAW-based pipelines, enabling broader deployment in edge-to-cloud scenarios. Therefore, evaluating methods based solely on downstream accuracy constitutes an unfair comparison, especially considering that RAW images typically require 48 bpp for storage and transmission.
>
> Following your suggestion, we have added two RAW-based methods, AdaptiveISP (*Learning an Adaptive Image Signal Processor for Object Detection*，NeurIPS2024) and SID (*Learning to See in the Dark*, CVPR 2018). Due to time and computational constraints, we are only able to report results on the YOLOv3 backbone.
>
> | | mAP | mAP50 | mAP75 |
> |---|---|---|---|
> | YOLOv3 | 40.00 | 67.67 |43.44  |
> | SID | 40.01| 68.69 |42.84  |
> | AdaptiveISP | 41.91 | 68.55 | 45.66 |
> | Ours | 42.14 | 69.11 | 46.02 |
>
> However, quite surprisingly, our method still outperforms these newly incorporated RAW based methods, which can be mainly attributed to the analytical design of our MV-IR module. It establishes a clear and efficient restoration path built upon prior knowledge of the ISP pipeline.Note that **RAWformer outputs RAW images**, which would require substantial modifications to downstream detectors; therefore, we do not include it in this comparison.We appreciate your understanding and will include more comprehensive comparisons with cutting-edge RAW-based methods in the revised manuscript.
>
>
>
> Q4. Compare the computational overhead and inference time.
>
> Ans.: Thanks for your valuable and constructive suggestions.  It is indeed important and necessary to analyze the running speed and computational overhead.
> As our method is mainly proposed for reducing the transmission cost in edge-to-cloud scenarios, where the source images (sRGB or RAW) also need to be compressed and transmitted, we benchmark both stages separately for a fair assessment of runtime and computational overhead. The resulting speed and computational-cost statistics are reported below.
>
> 1)Imaging end: We compare the CMCE encoder with extra two image codecs (Encoder part only): TIC (*Transformer-based Image Compression*, DCC 2022), and MLIC++ (*Linear-Complexity Multi-Reference Entropy Modelling*, arXiv 2023).;
>
> | | Para. (M) | Enc. Latency (ms) | GFLOPs |
> |---|---|---|---|
> | TIC | 12.74 | 3009.13 |338.59  |
> | MLIC++ |68.80  | 181.97 | 147.70 |
> | Ours (CMCE) | 1.78 | 40.72 | 38.17 |
>
> 2)Application end: we evaluate the MV-IR module against the five enhancement methods already employed in our manuscript.
>
> | | Para. (M) | Inf. Time (ms) | GFLOPs |
> |---|---|---|---|
> | Zero-DCE | 0.08 | 6.50 | 20.76 |
> | KinD | 8.02 | 5.23 | 258.91 |
> | YOLA | 0.01 | 2.00 | 2.32 |
> | RAOD | 0.07| 1.75 | 0.21 |
> | Ours (MV-IR) | 0.08 | 2.52 | 0.27 |

---

> > ### Comment · Reviewer_aGYu · 2025-08-05
> >
> > Thank you for your detailed rebuttal. Currently, I prefer to keep my rating of "4: Borderline accept" based on considering the overall contribution of this work.

---

### Official Review · Reviewer_rFZh · 2025-06-15

**Clarity:** 3
**Significance:** 3
**Originality:** 2
**Rating:** 4
**Confidence:** 4

**Summary:**

The authors present CRM-IR (Raw Metadata-guided Image Refinement), a novel raw-based object detection framework that achieves superior performance compared to conventional methods while excelling in flexibility and compactness.
To realize this, the authors designed two key components: CMCE and MV-IR, and introduced a dedicated evaluation dataset.

**Questions:**

Please refer to the weaknesses section. My current rating could change significantly depending on the authors’ responses. While I have some knowledge of camera ISP and image restoration, I do not have specialized expertise in the specific field of this study. I am aware of this and will actively refer to other reviewers’ questions and the authors’ responses as well.

**Ethical Concerns:**

["NO or VERY MINOR ethics concerns only"]

**Final Justification:**

The authors have addressed all of my concerns, and they have been fully resolved.

**Limitations:**

The authors do not mention any limitations of this study. I hope they will add this to Section 5. Conclusion, and I would like to hear their thoughts on this point in the rebuttal.

**Paper Formatting Concerns:**

No formatting issues have been found.

**Quality:**

2

**Strengths And Weaknesses:**

[Strengths]
1. The main motivation is simple, and reasonable.
2. The new evaluation dataset was provided.
3. The paper is well-organized and easy to read.

[Weaknesses]
1. It is questionable whether each module was trained as intended by the authors and whether this led to the observed performance improvements. For example, Section 4.3 only shows the performance degradation when CMCE and MV-IR are simply removed. Based on these experimental results, it is difficult to verify whether CMCE effectively captures the inter-redundancy between the raw and the sRGB images as intended by the authors, and whether the inverse mapping from the linear representation in MV-IR is being performed as intended.

2. The baseline models(e.g. YOLOv3(2018), Faster R-CNN(2015), CenterNet(2019)) used are outdated, and the performance trends vary significantly across baselines, raising doubts about whether the authors' proposed method would bring meaningful performance improvements in recent models. For example, in Table 3, when CenterNet(2019) is used as the backbone, the performance differences between YOLA, RAOD, and the proposed method(Ours) are not statistically significant. This trend is likely to become more pronounced when evaluated against newer models. Additionally, I am curious why there was no comparison with models such as AdaptiveISP (NeurIPS 2024).

3. This model is specialized for the ISP of the camera used to capture the training data, and its practical application assumes imaging with the target camera. In this context, measuring generalization ability on datasets captured with "different cameras" seems questionable. If the LOD dataset lacks sufficient extremely low-light conditions, collecting additional evaluation datasets should use the same camera to ensure meaningful comparisons.sets should involve the same camera to ensure meaningful comparisons.

4. I'm curious why camera parameters that directly influence the ISP processing, such as exposure time and ISO, were not utilized.

5. I am curious why the mAP75 performance of RAOD is particularly low except when using Faster R-CNN as the backbone.

6. In the table, all the bpp values for the comparison models are marked as "-", is this report accurate?

7. The authors do not mention any limitations of this study. If the authors have any limitations in mind, I would appreciate it if they could describe them in detail.

8. Minor error.
-- L142: y >> y_hat
-- L164: (redundant phrase)
-- L172: Shouldn't a reference be provided?
-- L266: YOLOA >> YOLA
-- All tables: map >> mAP

---

> ### Author Rebuttal · Authors · 2025-07-31
>
> First of all, we want to express our sincere appreciation for your responsible and thoughtful review. Your suggestions not only help improve the quality of our current manuscript but also inspire our future work. Below, we answer your questions point-by-point in the hope of addressing your concerns, and we welcome any further suggestions.
>
> Q1: Concerns about the effectiveness of the proposed modules and the ablation study settings.
>
> Ans.: Thank you for your valuable and insightful suggestions, which made us realize that our current ablation studies are insufficient to demonstrate the effectiveness of our design.
>
> For the ablation of CMCE, we conducted an additional study that replaces the referenced sRGB images with plain images (containing no information) at the encoding side, while keeping all other components identical. This eliminates cross-modal correspondence during RAW-metadata compression, and we retrained the entire framework on the YOLO-V3 backbone. The results are listed below:
>
> | | mAP | mAP50 | mAP75 | bpp |
> |---|---|---|---|---|
> | Ours(Full) | 42.14 | 69.11 | 46.02 | 4.88e-4|
> | Ablated | 39.40 | 66.50 | 43.39 |0.25 |
>
>
> As clearly shown, omitting the correspondence between the sRGB and RAW images greatly increases the transmission-resource requirement and results in a remarkable performance drop, confirming that CMCE successfully leverages cross-modal redundancy.
> As for MV-IR, we conducted comprehensive ablation studies by alternately disabling the remapping and inverse-mapping steps to demonstrate their effectiveness on the YOLOv3 backbone; the results are shown below.
>
> | | mAP | mAP50 | mAP75 | bpp |
> |---|---|---|---|---|
> | Ours (Full) | 42.14 | 69.11 | 46.02 |4.88e-4 |
> | $w/o$ inv. mapping | 40.59 | 66.17 | 44.22 | 8.57e-4|
> | $w/o$ remapping |40.85  | 67.67 | 43.93 | 6.60e-4|
>
> Thank you again for your insightful suggestions. We will incorporate these new ablation studies into the revised manuscript.
>
> Q2. Evaluate the effectiveness on recent detectors.
>
> Ans.: Thank you for your suggestions. In response, we have added the cutting-edge object detector YOLOv11 as a new baseline and compared it with other restoration-then-detection schemes. The corresponding results are listed below.
>
> | | mAP | mAP50 | mAP75 | bpp |
> |---|---|---|---|---|
> | YOLOv11 | 50.59 | 70.12 |54.82  | -|
> | Zero-DCE | 50.28 | 69.58 | 54.82 | -|
> | KinD | 48.70 | 68.47 |52.93  | -|
> | YOLA | 50.99 | 70.84 | 55.95 | -|
> | RAOD | 51.87 | 71.56 | 56.04 | -|
> | Ours | 52.05 | 71.22 | 56.37 | 7.25e-4 |
>
> The results show that our method still obtains remarkable advantages over these enhancement-then-detection methods on recent detectors. We will include additional results using other SOTA detectors in the supplementary materials of the revised manuscript, as this is essential for a comprehensive demonstration of our method’s effectiveness. Thanks again for your valuable suggestion.
>
> Q3. Comparison with recent enhancement models
>
> Ans.: Thanks a lot for your suggestions. Following your suggestion, we have added two RAW-based enhancement methods AdaptiveISP (*Learning an Adaptive Image Signal Processor for Object Detection*，NeurIPS 2024) and SID (*Learning to See in the Dark*, CVPR 2018) for a more comprehensive comparison. Owing to time and computational-resource constraints, we are able to report results only on the YOLOv3 backbone; these results are shown below. We appreciate your understanding.
>
> Herein, we’d like to emphasis that compared with exiting RAW-based methods, we are not **focused solely on achieving higher performance**, which would be unrealistic because those approaches have full access to the entire RAW data. Our primary goal is to minimize the transmission and storage requirements of RAW-based pipelines so they can be deployed widely in edge-to-cloud scenarios, given that RAW images typically require 48 bpp for storage and transmission.
>
> | | mAP | mAP50 | mAP75 |
> |---|---|---|---|
> | YOLOv3 | 40.00 | 67.67 |43.44  |
> | SID | 40.01| 68.69 |42.84  |
> | AdaptiveISP | 41.91 | 68.55 | 45.66 |
> | Ours | 42.14 | 69.11 | 46.02 |
>
> However, quite surprisingly, our method still outperforms these newly incorporated RAW based methods, which can be mainly attributed to the analytical design of our MV-IR module. It establishes a clear and efficient restoration path built upon prior knowledge of the ISP pipeline. Moreover, we will incorporate more recent RAW-based methods into our related-work section to better contribute to the community.
>
> Q4. Concerns about the generalization evaluation across different cameras and the validity of cross-camera comparisons.
>
> Ans.: Thanks a lot for your kind and valuable suggestions. Herein, we would first like to emphasize that both the camera we used and the one used for the LOD dataset are Canon cameras. Although they are different models, the RAW images were captured at the same format. Moreover, our model is designed to extract only the essential metadata from the information that is discarded by the ISP. Since most information loss in the ISP occurs during gamma correction and compression (processes that depend only minimally on the camera model) the MV-IR module applies an inverse-gamma operation followed by a learnable remapping to recover the information lost. From this perspective, we hold a positive attitude toward our model’s cross-camera generalization capability.
>
> Q5. Explanation for not utilizing exposure time and ISO
>
> Ans.: Thanks for your comments. First, we acknowledge that exposure time and ISO are important and are widely used in RAW reconstruction and low-light enhancement studies which pursues high reconstruction fidelity or perceptual quality. However, our method focuses on fulfilling machine-vision needs while minimizing transmission cost, as we apply no fidelity or perception-oriented supervision.
>
> Moreover, we would like to claim that with the access to both the sRGB images and the RAW counterpart, these parameters can be easily derived analytically, in other words, the obtained sRGB images have already contained such information.
>
> Accordingly, we do not employ these camera parameters directly; instead, we rely on end-to-end training to automatically extract the most essential metadata.
>
> Q6. Concerns regarding the unexpectedly low mAP75 performance of RAOD
>
> Ans.: Thanks a lot for your comments. According to our analysis, the discrepancy stems from the training preferences of different object detection models. Factors such as higher NMS thresholds, the choice of GIoU/DIoU/CIoU losses, and bounding box regression bias can influence mAP50 and mAP75 differently. Since most detectors prioritize the mAP50 metric, their hyperparameters are often tuned to favor it. As a result, performance comparisons between mAP50 and mAP75 may exhibit different trends. Moreover, we strictly adhere to the principle of reproducibility and will open-source all checkpoints and training scripts for the baseline models.
>
> Q7. Concerns regarding the accuracy of the reported bpp values
>
> Ans.: Thanks for your kind comments. We apologize for any confusion caused by our unclear representation. In Tables 1–3 of our manuscript, the bpp value accounts only for the additional bits required to transmit the RAW metadata. In fact, as noted in lines 263 and 272 of our manuscript, the bpp values for the sRGB and RAW images are 0.15 and 48, respectively. Therefore, the total cost in our setting is 0.15 (sRGB)+ 4.88e-4(RAW Metadata). We will revise the manuscript to make this point clearer.
>
>  Moreover, the bpp values in our manuscript are measured at test time and accurately represent the actual transmission cost in practical deployments. During inference, the RAW metadata generated by the CMCE module is coded into a binary bit-stream and transmitted to the application end. Its bpp is calculated as the bit-stream size (in bits) divided by the image resolution. The bpp for both the sRGB and RAW reference images is computed in the same way, using each file’s size (in bits) divided by its resolution.
>
>  Q8. Conclude the limitations of this work
>
> Ans.: Thanks a lot for your kind suggestion. We ‘d like to conclude our limitations as follows: First, as mentioned in Sec. 3.4, our RID dataset is an ongoing project. Inspired by your suggestions, we believe it is necessary to expand the dataset by incorporating images captured from different cameras and smartphones, and to further investigate the variation in RAW metadata requirements across different ISP pipelines. Second, our MV-IR module is currently optimized for object detection, and its effectiveness for other machine vision tasks (e.g., semantic segmentation) remains unexplored. Creating corresponding annotations and either tailoring the module for specific tasks or developing a generalized version are valuable directions for future work. Third, inspired by your comments, we find it interesting to investigate the differences in required metadata between machine vision needs and human perception–oriented needs. Furthermore, designing a hierarchical structure to represent both machine- and perception-tailored metadata, and incorporating a scalable compression strategy to fulfill both requirements, would be an interesting and valuable direction for extending our current work.
>
> Q9. Minor issues related to typos and missing references
>
> Ans.: Finally, we would like to express our deepest gratitude for your comprehensive and responsible review, which has greatly helped improve the quality of our manuscript. We will revise the relevant sections accordingly, incorporate the necessary references, and conduct thorough proofreading to avoid such issues.

---

> > ### Comment · Reviewer_rFZh · 2025-08-03
> >
> > First of all, I truly appreciate the effort you put into conducting additional analyses within a short time frame and providing a thoughtful response. Most of my questions have been satisfactorily addressed. I do, however, have a few additional inquiries:
> >
> > For [Q1], in the first experiment, would it be possible to examine the performance when the sRGB image is either removed or replaced with a plain image instead of using the original sRGB?
> > In the second experiment, could you elaborate in more detail on how the metadata and sRGB are handled in the "w/o inv. mapping" and "w/o remapping" settings?

---

> > > ### Author Response · Authors · 2025-08-04
> > > **Response to reviewer rFZH**
> > >
> > > We are very glad and encouraged to hear that our responses have addressed most of your questions. Thanks again for your kind suggestions and your positive attitude toward our work. Regarding your remaining questions, we hope the following answers can adequately address them.
> > >
> > > Q1. Examine the performance when remove the original sRGB input.
> > >
> > > Ans.: Thanks a lot for your kind suggestions.
> > >
> > > Sorry for that we previously did not express this clearly. In both CMCE and MV-IR, sRGB images are involved. In our first-round response, we have already replaced the sRGB images with plain images (all pixels set to 0) in the CMCE module, meaning that no RAW–sRGB cross-modal correspondence was leveraged for metadata compression. However, at the MV IR (enhancement) stage, the sRGB image was still used as input.
> > >
> > > Following your suggestion, we further replaced the sRGB images with plain images as the input to the MV-IR module, i.e., the entire pipeline operates without access to the sRGB image.
> > > A more comprehensive ablation results are listed below (including ones presented in previous response):
> > >
> > >
> > >
> > > | | CMCE input | MV-IR input | mAP50 | bpp |
> > > |---|---|---|---|---|
> > > | Full (Ori. version) | sRGB+RAW | sRGB | 69.11 | 4.88e-4|
> > > | $1^{\text{st}}$ round version | Plain (all 0)+RAW | sRGB | 66.50 |0.25 |
> > > | Current version | Plain (all 0)+RAW | Plain (all 0) | 0 |1.36e-4 |
> > >
> > > As shown, completely removing the sRGB images leads to a significant catastrophic performance drop. The MV-IR module, designed for image refinement with only 0.08M parameters, produces essentially plain images instead of full reconstructions, preventing the CMCE module from extracting meaningful RAW metadata. As a result, this leads to a significant drop in bpp loss and negligible bpp requirements during testing.
> > >
> > >
> > > Q2. Elaborate in more detail on how the metadata and sRGB are handled in the "$w/o$ inv. mapping" and "$w/o$ remapping" settings
> > >
> > > Ans.: Thank you for your suggestions. As shown in Fig. 2(b) and (c) of our manuscript, the inverse mapping and remapping processes incorporate dedicated layers to process the RAW metadata, with the aim of estimating image-wise gamma correction parameters and pixel-wise modulation maps, respectively.
> > >
> > > Herein, for the "$w/o$ remapping" setting, the inverse gamma–corrected image (i.e., $x_l$ in Fig. 2(b)) is treated as the final output and directly fed to the downstream detector.
> > >
> > > As for the "$w/o$ inv. mapping" setting, the learned pixel modulation map $\mathbf{m}$ is directly added to the input sRGB image instead of the inverse gamma–corrected image $x_l$, while keeping all other layers unchanged.

---

> > > > ### Comment · Reviewer_rFZh · 2025-08-04
> > > >
> > > > Among the follow-up questions I asked, the second one has been resolved thanks to the authors’ kind explanation.
> > > > There was a misunderstanding regarding the first question. I sincerely apologize for the confusion.
> > > >
> > > > Regarding the first experiment in [Q1], I was actually wondering whether it would be possible to evaluate the performance when the RAW image, rather than the sRGB image, is either removed or replaced with a plain image.
> > > > If available, would it be possible for you to share the results of such a case?

---

> > > > > ### Author Response · Authors · 2025-08-05
> > > > > **Response to reviewer rFZH**
> > > > >
> > > > > Q1. Examine the performance when replacing the RAW images with plain images.
> > > > >
> > > > > Ans.: Thanks for your kind suggestions.
> > > > >
> > > > > Herein, we replace the RAW images with plain images, retrain the entire framework. The corresponding results on the Yolov3 backbone is listed below.
> > > > >
> > > > > | | mAP | mAP50 | mAP75 | bpp |
> > > > > |---|---|---|---|---|
> > > > > | Ours(Full) | 42.14 | 69.11 | 46.02 | 4.88e-4|
> > > > > | $w/o$ RAW | 40.74 | 67.47 | 44.30 |8.41e-6 |
> > > > >
> > > > > As shown, when the RAW images are removed, the results closely resemble those from the ablation study in our manuscript where the entire CMCE module was ablated. Moreover, with access only to the sRGB images, the CMCE module is unable to extract any meaningful information, resulting in a negligible bpp requirement during testing.

---

> > > > > > ### Comment · Reviewer_rFZh · 2025-08-05
> > > > > >
> > > > > > I sincerely appreciate the authors’ kind and thorough responses until the very end. All of my questions have been fully addressed, and I now believe the manuscript has sufficient merit to warrant an upgrade in rating. Thank you for your hard work.

---

### Official Review · Reviewer_VyZs · 2025-07-01

**Clarity:** 3
**Significance:** 2
**Originality:** 3
**Rating:** 5
**Confidence:** 4

**Summary:**

This work proposed an approach for low-light image enhancement by partially using the raw sensor data.
It estimates the metadata from the raw image using a trainable probabilistic encoder, then uses the metadata to remap the sRGB image to a domain optimal for machine vision tasks. Main novelties include:

1. An entropy parameter network for the metadata extraction
2. A selective remapping technique that treats the low-light part and the high-light part of the image separately.

**Questions:**

1. The ablation study is done on the latest generations of detection models.  YOLA and RAOD, which are compared in Tables and 3, are not used in the ablation study. Can the authors explain the reason of it?

**Ethical Concerns:**

["NO or VERY MINOR ethics concerns only"]

**Final Justification:**

As the authors explained my questions regarding the metric comparison, I am willing to recommend this paper as accept.

**Paper Formatting Concerns:**

Syntax in line 164.

**Quality:**

3

**Strengths And Weaknesses:**

Strength:
1. This work uses metadata for the image enhancement, which is less computationally expensive than those that directly use the raw image.
2. This work does not require a trainable ISP for the raw data, thus it is expandable to different schemes.

Weakness:
1. The estimation of bpp is not stated clearly. Is it purely the transmission and storage overhead for the metadata? If so, the computational cost of the encoder is not counted. A proper comparison of the total FLOPs against other models might be useful.

---

> ### Author Rebuttal · Authors · 2025-07-31
>
> First of all, we would like to express our deepest gratitude for your positive attitude and valuable suggestions, which are very important and a great encouragement to us. We hope that our following responses adequately address your concerns.
>
> Q1. Explain the bpp calculation.
>
> Ans.: Thanks for your thoughtful review, and we apologize for any confusion caused by the missing details. The bpp values in our manuscript are measured at test time and accurately represent the actual transmission cost in practical deployments. During inference, the RAW metadata generated by the CMCE module is coded into a binary bit-stream and transmitted to the application end. Its bpp is calculated as the bit-stream size (in bits) divided by the image resolution. The bpp for both the sRGB and RAW reference images is computed in the same way, using each file’s size (in bits) divided by its resolution.
>
> Q2. Compare the computational cost against other models.
>
> Ans.: Thanks for your valuable and constructive suggestions.  It is indeed important and necessary to analyze the running speed and computational overhead, which are critical in real-world applications.
> As our method is mainly proposed for reducing the transmission cost in edge-to-cloud scenarios, where the source images (sRGB or RAW) also need to be compressed and transmitted, we benchmark both stages separately for a fair assessment of runtime and computational overhead.All GPU experiments run on an NVIDIA RTX 6000. The resulting speed and computational-cost statistics are reported below.
>
> 1)Imaging end: We compare the CMCE encoder with extra two image codecs (Encoder part only): TIC (*Transformer-based Image Compression*, DCC 2022), and MLIC++ (*Linear-Complexity Multi-Reference Entropy Modelling*, arXiv 2023).;
>
> | | Para. (M) | Enc. Latency (ms) | GFLOPs |
> |---|---|---|---|
> | TIC | 12.74 | 3009.13 |338.59  |
> | MLIC++ |68.80  | 181.97 | 147.70 |
> | Ours (CMCE) | 1.78 | 40.72 | 38.17 |
>
> 2)Application end: we evaluate the MV-IR module against the five enhancement methods already employed in our manuscript.
>
> | | Para. (M) | Inf. Time (ms) | GFLOPs |
> |---|---|---|---|
> | Zero-DCE | 0.08 | 6.50 | 20.76 |
> | KinD | 8.02 | 5.23 | 258.91 |
> | YOLA | 0.01 | 2.00 | 2.32 |
> | RAOD | 0.07| 1.75 | 0.21 |
> | Ours (MV-IR) | 0.08 | 2.52 | 0.27 |
>
> Q3. Concerns of the absence of YOLA and RAOD in ablation studies.
>
> Ans.: Thanks for your kind suggestions. We apologize for any misunderstanding caused by our unclear representation. To address your concern, we would like to clarify that in Tables 1–3 we compare the commonly employed **restoration-then-detection** paradigm: that first apply a low-light enhancement model to obtain high-quality images, then feed these images to downstream object-detection models. In fact, both YOLA and RAOD are machine-vision-tailored low-light enhancement methods; thus, we did not employ them in the ablation study. After carefully proofreading the manuscript, we realized that our statement in line 235 may mislead readers, and we will modify the corresponding part in the revised manuscript to avoid this issue.

---

> > ### Comment · Reviewer_VyZs · 2025-08-04
> >
> > I thank the authors for the clarification. I still have one more question regarding the computational efficiency.
> >
> > As the authors compared the proposed model GFLOPs against MLIC++ and TIC, I am also curious about the introduced bpp of these two encoders in different image processing backbones. Could the authors provide this?

---

> > > ### Author Response · Authors · 2025-08-05
> > > **Response to reviewer VyZs**
> > >
> > > Thanks again for your kind suggestions and your support of our work, which is sincerely appreciated. We hope our following answer would adequately address your left question.
> > >
> > > Q1. Examine the introduced bpp of MLIC++ and TIC in different image processing backbones.
> > >
> > > Ans.: Thanks for your kind suggestions. We sincerely appreciate your insightful comments, which have helped enhance the completeness and rigor of our work.
> > >
> > > As the sRGB images used in our study are already JPEG-compressed, with an average bpp of approximately 0.15, it is difficult for us to conduct a fair and meaningful analysis under uncompressed conditions. Unfortunately, the dataset does not provide access to the original, pristine sRGB images.
> > >
> > > However, based on our knowledge of image compression domain, cutting-edge codecs such as MLIC++ and TIC generally achieve a 50%–100% improvement in compression performance compared to JPEG. Therefore, under fair comparison settings, we estimate that using MLIC++ or TIC would result in a bpp of approximately 0.1. We hope this estimate is helpful, and we would greatly welcome any further suggestions you might have on this topic.
> > >
> > > Furthermore, inspired by your comment, we recognize that our framework has the potential to be extended with more compression backbones. Our MECE module is inherently compatible with modern learned image codecs and can be flexibly integrated with advanced encoder architectures and entropy models. We plan to explore this direction further in our future work.
> > >
> > > As for RAW images, they are typically treated as lossless formats and require lossless compression (48 bpp) when used in RAW-based methods. However, common image codecs are lossy and may cause significant information loss, thereby compromising the unique advantages of RAW data. As such, we list the 48 bpp of the transmission cost of the RAW images in our manuscript.

---

### Official Review · Reviewer_4Xo9 · 2025-07-02

**Clarity:** 2
**Significance:** 2
**Originality:** 2
**Rating:** 4
**Confidence:** 4

**Summary:**

A novel low-light object detection paradigm, Compact RAW Metadata-guided Image Refinement (CRM-IR), is proposed. The core idea of this method is to avoid directly using RAW images for inference, and instead, leverage compressed RAW metadata to guide the enhancement of sRGB images. This enables improved object detection performance under low-light conditions without modifying existing detection models or the camera's ISP pipeline.

**Questions:**

How does the metadata-guided inverse gamma and remapping in MV-IR fundamentally differ from traditional enhancement methods like Retinex?  Does it truly capture representations more suitable for machine vision?

The CMCE achieves extremely low bitrate (<0.001 bpp).  What specific cross-modal redundancies does it exploit, and how does it improve upon standard learned image compression techniques?

The evaluation primarily uses YOLOv3, Faster R-CNN, and CenterNet as baselines, which are relatively outdated compared to recent state-of-the-art detectors such as YOLOv11/v12,. This may limit the assessment of the method’s effectiveness in modern real-world deployment scenarios.

The paper does not analyze the runtime or computational overhead introduced by the MV-IR and CMCE modules.

**Ethical Concerns:**

["NO or VERY MINOR ethics concerns only"]

**Final Justification:**

Thank authors for the rebuttal. The authors have addressed most of my concerns. However, the current method still shows some limitations in enhancing sRGB images. Nevertheless, considering the valuable contribution to RAWs-based low-light enhancement, I am willing to raise my rating to a positive recommendation.

**Limitations:**

yes

**Quality:**

2

**Strengths And Weaknesses:**

Quality: The authors are based on raw and SRGB pairs of low light, but there seem to be limitations in the experimental results (many indicators are worse than YOLA RAOD) .

Clarity: Low-Light Object Detection combined with Metadata from RAWs is natural and well-motivated, and the paper presents a clearly structured framework.

Significance: The method requires no modifications to the ISP or downstream detection model architectures, making it easy to integrate into existing vision pipelines. It achieves comparable object detection accuracy while significantly reducing RAW data storage and transmission costs by transmitting only compact metadata instead of full-resolution RAW images.

Originality: The novelty of this approach appears limited, as its performance gains seem inferior to both YOLA (sRGB-based) and RAOD (raw-based) methods. This raises legitimate doubts about whether the cross-modal design actually delivers meaningful improvements.

---

> ### Author Rebuttal · Authors · 2025-07-31
>
> First of all, we would like to express our sincere appreciation for your responsible and comprehensive review. Your positive attitude toward our RAW-metadata-guided, plug-and-play design is a great encouragement to us. Regarding your concerns, we hope the following point-by-point responses address them, and we would appreciate any further suggestions.
>
> Q1. Concerns about the fundamental differences from traditional enhancement methods.
>
> Ans.: Thanks for your comments. First, we would like to emphasize that, compared with traditional Retinex-based methods whose heuristic illumination estimation is agnostic to camera settings and recovers only approximate reflectance, our approach begins in the RAW domain to reconstruct the linear sensor-radiance domain. The inverse-gamma and remapping operations are derived analytically from the RAW images, making the mapping physically interpretable.
>
> Besides our work, recent RAW-based low-light-enhancement methods have become mainstream and demonstrate overwhelming advantages over traditional approaches. Compared with these RAW-based methods, we focus on practical edge-to-cloud scenarios, where RAW images cannot be directly accessed, and aim to minimize transmission cost by identifying and extracting only the most essential RAW data.
>
> For intuitive demonstration, we include two Retinex-based methods, one conventional method LIME (*LIME: Low-Light Image Enhancement via Illumination Map Estimation*，TIP 2016) and one learning-based method RetinexFormer (*Retinexformer: One-stage Retinex-based Transformer for Low-light Image Enhancement*, ICCV 2023). We first compare their performance with ours on the LOD dataset, then feed their outputs into our pipeline for additional RAW-metadata-guided enhancement; the corresponding results are shown below.
>
> | | Ours | LIME | RetinexFormer | LIME+OURS | RetinexFormer+OURS |
> |---|---|---|---|---|---|
> | mAP50 | 69.11|67.56 |65.95 | 68.91|67.77 |
> | bpp | 4.88e-4 | - | - | 4.92e-4 | 4.77e-4 |
>
> As shown, our method outperforms the conventional methods and can further enhance their performance, demonstrating that RAW metadata offers remarkable advantages over these approaches.
>
> Q2. Explain the specifically leveraged specific cross-modal redundancies and improvement upon standard learned image compression.
>
> Ans.: Thanks for your comments. Cross-modal redundancy denotes the contextual correspondence between a RAW image and its compressed sRGB counterpart, both of which are available at the imaging end. Because the sRGB image is a degraded yet closely correlated representation of the same scene, the two modalities share substantial mutual information. This allows our method to transmit only a small amount of RAW-specific data, since the decoder already possesses the sRGB frame. In contrast to standard learned image codecs, which handle a single modality and remove only intra-image redundancy, we take as input both RAW and sRGB and improve the entropy model to exploit their correspondence, thereby further reducing the required bitrate.
>
> Moreover, our method does not aim for signal-level fidelity, meaning it does not reconstruct the RAW image on the decoding side. The end-to-end strategy uses compactness and task-specific performance as its constraints, so it extracts only the key information needed by downstream tasks. These task-oriented adaptations, compared with existing learned image codecs, enable an extremely low bitrate.
>
> Q3. Evaluate the effectiveness on recent detectors.
>
> Ans.: Thanks a lot for your kind suggestion. Due to time and computational-resource limitations, we were only capable of adopting only YOLOv11 (-x version) as the extra benchmark, we hope your understanding. Comparisons were conducted on the LOD dataset, with all settings kept exactly the same as in our manuscript for fair comparison; the results are shown below.
>
> | | mAP | mAP50 | mAP75 | bpp |
> |---|---|---|---|---|
> | YOLOv11 | 50.59 | 70.12 |54.82  | -|
> | Zero-DCE | 50.28 | 69.58 | 54.82 | -|
> | KinD | 48.70 | 68.47 |52.93  | -|
> | YOLA | 50.99 | 70.84 | 55.95 | -|
> | RAOD | 51.87 | 71.56 | 56.04 | -|
> | Ours | 52.05 | 71.22 | 56.37 | 7.25e-4 |
>
> The results show that our method still obtains remarkable advantages over these enhancement-then-detection methods on recent detectors. We will include additional results using other SOTA detectors in the supplementary materials of the revised manuscript, as this is essential for a comprehensive demonstration of our method’s effectiveness. Thanks again for your valuable suggestion.
>
> Q4. Analyze the runtime and computational overhead.
>
> Ans.: Thanks for your valuable suggestions, it is indeed important and necessary to analyze the running speed and computational overhead for reference.
>
> As our method is mainly proposed for reducing the transmission cost in edge-to-cloud scenarios, where the source images (sRGB or RAW) also need to be compressed and transmitted, we benchmark both stages separately for a fair assessment of runtime and computational overhead.All GPU experiments run on an NVIDIA RTX 6000. The resulting speed and computational-cost statistics are reported below.
>
> 1)Imaging end: We compare the CMCE encoder with extra two image codecs (Encoder part only): TIC (*Transformer-based Image Compression*, DCC 2022), and MLIC++ (*Linear-Complexity Multi-Reference Entropy Modelling*, arXiv 2023).;
>
> | | Para. (M) | Enc. Latency (ms) | GFLOPs |
> |---|---|---|---|
> | TIC | 12.74 | 3009.13 |338.59  |
> | MLIC++ |68.80  | 181.97 | 147.70 |
> | Ours (CMCE) | 1.78 | 40.72 | 38.17 |
>
> 2)Application end: we evaluate the MV-IR module against the five enhancement methods already employed in our manuscript.
>
> | | Para. (M) | Inf. Time (ms) | GFLOPs |
> |---|---|---|---|
> | Zero-DCE | 0.08 | 6.50 | 20.76 |
> | KinD | 8.02 | 5.23 | 258.91 |
> | YOLA | 0.01 | 2.00 | 2.32 |
> | RAOD | 0.07| 1.75 | 0.21 |
> | Ours (MV-IR) | 0.08 | 2.52 | 0.27 |
>
> Q5. Concerns about the performance gains to both YOLA (sRGB-based) and RAOD (raw-based) methods.
>
> Ans.: Thanks for your comments. 1)As for for the sRGB-based method YOLA, we outperform it in 17 of 18 comparison groups (2 datasets $\times$ 3 backbones $\times$ 3 metrics); notably, in cross-dataset testing our method achieves a 4 % mAP50 gain across all three backbone detectors, clearly demonstrating the effectiveness of leveraging RAW information.
>
> Q6. Moreover, compared with RAOD and other RAW-based methods, we are not focused solely on achieving higher performance, which would be unrealistic because those approaches have full access to the entire RAW data. Our primary goal is to minimize the transmission and storage requirements of RAW-based pipelines so they can be deployed widely in edge-to-cloud scenarios. Hence, evaluating only downstream accuracy is an unfair comparison, given that RAW images typically require 48 bpp for storage and transmission.

---

> > ### Comment · Reviewer_4Xo9 · 2025-08-04
> >
> > Thank authors for the rebuttal. The authors have addressed most of my concerns. While the current method still shows some limitations in enhancing sRGB images, considering the valuable contribution to RAWs-based low-light enhancement, I am willing to raise my rating to a positive recommendation.

---

### Decision · Program_Chairs · 2025-09-17

**Decision:**

Accept (poster)

**Comment:**

This paper proposes a new RAW-based machine vision framework called Compact RAW Metadata-guided Image Refinement (CRM-IR). In particular, the paper proposes a Machine Vision-oriented Image Refinement (MV-IR) module that refines sRGB images into a format suitable for machine vision based on learned RAW metadata.


The proposed method does not require any changes to ISP or downstream detection model architectures, enabling easy integration into existing vision pipelines. Additionally, by transmitting only compact metadata rather than full-resolution RAW images, it significantly reduces RAW data storage and transmission costs while maintaining equivalent object detection accuracy.


Reviewers pointed out that the fundamental differences from traditional methods are unclear. They also noted that the methodological positioning, effectiveness, and utilization of cross-modal redundancy are unclear. Additionally, the effectiveness of the method for recent detectors has not been sufficiently evaluated. A detailed analysis of execution time and computational overhead is lacking. Furthermore, ablation experiments including YOLA and RAOD have not been conducted to ensure a fair comparison. Finally, the research contribution of the constructed dataset has not been sufficiently demonstrated.


The authors appropriately addressed the concerns, and all reviewers are satisfied with their responses. The paper received positive scores from all reviewers. Therefore, the AC believes that this paper provides the NeurIPS community as a whole with useful information. However, the AC strongly recommends incorporating the reviewers' comments into the camera-ready version.